# Identification of *JPX-RABEP1* Pair as an Immune-Related Biomarker and Therapeutic Target in Pulmonary Arterial Hypertension by Bioinformatics and Experimental Analyses

**DOI:** 10.3390/ijms232415559

**Published:** 2022-12-08

**Authors:** Qian Gong, Zhewei Hu, Qiao Jin, Yan Yan, Yan Liu, Jin He, Lenan Zhuang, Huanan Wang

**Affiliations:** 1Department of Veterinary Medicine, College of Animal Sciences, Zhejiang University, Hangzhou 310058, China; 2Key Laboratory of Cardiovascular Intervention and Regenerative Medicine of Zhejiang Province, Department of Cardiology, Sir Run Run Shaw Hospital, College of Medicine, Zhejiang University, Hangzhou 310016, China; 3Institute of Genetics and Reproduction, College of Animal Sciences, Zhejiang University, Hangzhou 310058, China

**Keywords:** bioinformatics analysis, pulmonary arterial hypertension, non-coding RNAs, immune cell infiltration

## Abstract

Pulmonary arterial hypertension (PAH) is a pulmonary vascular disease characterized by pulmonary vascular remodeling and right heart enlargement the pathogenesis of PAH is complicated; no biologic-based therapy is available for the treatment of PAH, but recent studies suggest that inflammatory response and abnormal proliferation of pulmonary artery smooth muscle cells are the main pathogenic mechanism, while the role of immune-related long non-coding RNAs (lncRNAs) remains unclear. The aim of this study was to systematically analyze immune-related lncRNAs in PAH. Here, we downloaded a publicly available microarray data from PAH and control patients (GSE113439). A total of 243 up-regulated and 203 down-regulated differentially expressed genes (DEGs) were screened, and immune-related DEGs were further obtained from ImmPort. The immune-related lncRNAs were obtained by co-expression analysis of immune-related mRNAs. Then, immune-related lncRNAs-mRNAs network including 2 lncRNAs and 6 mRNAs was constructed which share regulatory miRNAs and have significant correlation. Among the lncRNA-mRNA pairs, one pair (*JPX-RABEP1*) was verified in the validating dataset GSE53408 and PAH mouse model. Furthermore, the immune cell infiltration analysis of the GSE113439 dataset revealed that the *JPX-RABEP1* pair may participate in the occurrence and development of PAH through immune cell infiltration. Together, our findings reveal that the lncRNA-mRNA pair *JPX-RABEP1* may be a novel biomarker and therapeutic target for PAH.

## 1. Introduction

Pulmonary arterial hypertension (PAH) is a kind of progressive vascular disease characterized by medial hypertrophy and hyperplasia of the pulmonary artery. Ten percent of individuals with PAH are over 65, while the prevalence of the disease is estimated to be 15% worldwide. [1]. An increasing amount of evidence connects immunological infiltration, including T and B lymphocytes, macrophages, dendritic cells and mast cells, to PAH pathogenesis, illuminating the complicated pathophysiology [2]. Most patients with PAH have right heart dysfunction or even sudden death before they are detected [3]. Therefore, early diagnosis, early intervention and prognosis assessment are of great clinical significance to improve the survival rate, prognosis and precise treatment of PAH patients.

Long non-coding RNAs (lncRNAs) are a group of non-coding RNAs (ncRNAs) whose transcripts are more than 200 nucleotides in length [4]. It has been reported that lncRNAs are involved in the occurrence of almost all biological processes, including epigenetic regulation [5], translational regulation [6], transcriptional regulation [7] and post-transcriptional regulation [7]. Several studies have shown that lncRNAs can be involved in cardiovascular disease [8,9,10], and thus have been proposed as new targets for drug intervention. In recent years, the mechanism of lncRNAs in PAH has been widely studied [11,12,13]. However, the differential expression of immune-related lncRNAs, pathophysiological functions, and potential interactions of immune-related lncRNAs and mRNAs in PAH are still largely unknown.

With the development of whole transcriptome analyses, bioinformatics methods can be used to mine gene chip data to screen out the differentially expressed genes quickly and effectively. They are widely used in elucidating the pathogenic mechanism of diseases and the precise diagnosis and screening of drug therapeutic targets [14]. In our study, we investigated the expression profiles of mRNAs and lncRNAs in the lung tissues by downloading a publicly available microarray dataset from 11 control and 15 PAH patients (GSE113439) [15]. We identified immune-related lncRNAs using the competing endogenous RNA (ceRNA) network and constructed an ceRNA network containing 2 immune-related lncRNAs and six immune-related mRNAs. Finally, a pair of lncRNA-mRNAs (*JPX-RABEP1*) was verified by the validation dataset GSE53408 [16], including 11 control and 12 PAH patients downloaded from the gene expression omnibus (GEO) dataset and mouse model. Together, our results showed that *JPX* may regulate *RABEP1* through miRNA (miR-145, miR-146ac, miR-216a, miR-216b, miR-24, miR-33ab, miR-129-5p), affecting CD8^+^ T cells, NK cells and eosinophils and other immune cells, and finally participating in the immune process of PAH.

## 2. Results

### 2.1. Differentially Expressed Genes (DEGs) between PAH and Control Lung Samples from GSE113439

The GSE113439 dataset contains lung samples from 11 controls and 15 PAH patients. The sample distributions were illustrated in PCA plots (Figure 1A). In the PCA plot, it can be seen that patients in the control and PAH groups are clustered in separate circles. The differentially expressed genes (DEGs) in the GSE113439 dataset were screened with *p*-values < 0.05 and |Log2 fold change| > 1 as the empirical analysis cutoff, 446 DEGs were shown by the volcano plot (Figure 1B), of which 243 were upregulated and 203 were downregulated. The heatmap of 446 DEGs are presented in Figure 1C.

### 2.2. Identification of Immune-Related mRNAs/lncRNAs and the Construction of Immune-Related ceRNA Network

ceRNA is a novel molecular regulation mechanism in which non-coding RNAs and protein-coding genes compete for miRNA binding via miRNA response elements (MREs) [17]. lncRNAs can act as miRNA sponges to adsorb miRNAs and play the role of ceRNAs, affecting the function of miRNAs, thereby regulating the expression of mRNAs, and ultimately regulating various biological processes [18]. The expression levels of lncRNA and mRNA are positively correlated in the ceRNA network. The involvement of inflammation in the onset of the illness has recently come to light in preclinical and clinical PAH investigations. It was first noted that several inflammatory disorders, such as connective tissue illnesses, are linked to an elevated prevalence of PAH. Then, in lung samples from PAH patients, nearly all lineages of inflammatory cells—most notably macrophages, mast cells, T lymphocytes, B lymphocytes, dendritic cells, and neutrophils—were found close to the altered pulmonary vasculature [19]. Therefore, a list of immune-related genes was downloaded from the gene list resources in the Immunology Database and Analysis Portal (ImmPort) (https://www.immport.org/ (accessed on 1 June 2022)). Since there is only mRNA in the immune-related gene set, a total of 199 immune-related mRNA were identified in the DEGs (Appendix A).

The intersection of the above immune-related gene set and mRNAs of the DEGs were obtained and defined as immune-related mRNAs. The immune-related lncRNAs were screened by co-expression analysis of immune-related mRNAs with the |Pearson’s correlation coefficient| > 0.8. Based on the predicted validated miRNA-mRNA/lncRNA regulation and its associated levels, we established an immune-related ceRNA network. In this network, immune-related lncRNAs and mRNAs are predicted by shared miRNAs. The miRcode database was used to collect predicted and experimentally validated miRNA-mRNA-interaction data, as well as miRNA-lncRNA-interaction data. The screening criteria for lncRNA-mRNA pairs used to construct the ceRNA network are as follows: (1) the lncRNA and mRNA must share at least one miRNA; and (2) lncRNA and mRNA expression must be positively correlated (|Pearson’s correlation| > 0.8). The lncRNAs-miRNAs pairs in the co-expression analysis are shown in Appendix A for a total of 132 pairs, the mRNAs-miRNAs pairs in the co-expression analysis are shown in Appendix A for total 2404 pairs, and a total of 203 lncRNA-mRNA pairs were obtained (Appendix A). To make the obtained results more convincing, we increased the cutoff with |correlation co-efficiency| > 0.91, and the results were listed in Table 1.

The results showed that two lncRNAs and six mRNAs with shared miRNAs are included in the network (Figure 2A). Subsequently, we validated the lncRNAs and mRNAs screened in the ceRNA network in the training dataset GSE113439. The results showed that 6 mRNAs (Figure 2B) and 2 lncRNAs (Figure 2C) were significantly upregulated in PAH (*p* value < 0.0001).

### 2.3. Validation of the Immune-Related lncRNA-mRNA Pairs

To confirm the accuracy of the obtained results, we used another PAH microarray dataset GSE53408, including 11 control and 12 PAH patients as the validation dataset. First, two lncRNAs and six mRNAs were detected in the validation dataset GSE53408. Among these mRNAs (Figure 3A) and lncRNAs (Figure 3B), all lncRNA-mRNA pairs (*JPX-RABEP1*, *JPX-IREB2*, *MALAT1-CHUK*, *MALAT1-TANK*, *MALAT1-ECD* and *MALAT1-TBK1*) included in the ceRNA network were significantly upregulated in PAH.

These 2 lncRNAs and 6 mRNAs were consistent in both the training dataset GSE113439 and the validation dataset GSE53408. We then verified the correlation of these 6 lncRNAs-mRNAs pairs in these two datasets separately. The results showed that all pairs are positively correlated, among the training dataset GSE113439 (Figure 4A–F) and validation datasets GSE53408 (Figure 4G–L), respectively.

### 2.4. Validation of the Immune-Related lncRNA-mRNA Pairs in an Animal Model

To further verify the reliability of the obtained lncRNA-mRNA pairs in PAH, lung and heart samples were collected from a mouse model of PAH (gifted by Dr. Tingting Jin). First, we collected lungs from control and PAH mice for H&E staining (Figure 5A,B). The experimental results showed that compared with the control group, the alveolar volume of the PAH group increased, and the alveolar damage was severe. At the same time, it is accompanied by a thickening of the pulmonary arteriole wall, lumen stenosis, and proliferation of vascular smooth muscle cells. The arrows indicate the thickening of the pulmonary arteriole wall, lumen stenosis, and proliferation of vascular smooth muscle cells. These results indicated that the model of PAH was successfully established.

Since the development of PAH may lead to heart failure in the right ventricle, the lung tissues (Figure 6A–H) and heart tissues (Figure 6I–P) of mice were collected for qPCR to verify the expression levels of lncRNA-mRNA pairs. The expression levels of one pair of *JPX-RABEP1* were consistent with the results of the training and validating datasets.

### 2.5. Immune Cell Infiltration Analysis

To predict the immune cells infiltration between patients with PAH and the control group, we further performed the CIBERSORT algorithm by using the GSE113439 dataset. The percentage of each of the 22 types of immune cells in each sample was shown in the bar plot and heat map (Figure 7A,B). In the histogram, colors indicate the percentage of different immune cells in each sample, summed to 1. In the heatmap, immune cells in each sample are shown as normalized absolute abundance. The results indicated that resting memory CD4^+^ T cells, CD8^+^ T cells, resting mast cells, plasma cells, M2 macrophages, neutrophils, resting NK cells, monocytes, M0 macrophages and activated dendritic cells were the main infiltrating immune cells. The correlation of 22 types of immune cells in PAH lung tissues was then evaluated (Figure 7C). For example, activated memory CD4^+^ T cells, eosinophils and neutrophils were up-regulated in the PAH group, while CD8^+^ T cells, activated NK cells and resting mast cells were downregulated.

## 3. Discussion

In this study, we performed an integrated bioinformatics analysis of the clinical PAH microarray dataset GSE113439. To the best of our knowledge, this study is the first to identify the immune-related lncRNA-mRNA pair *JPX-RABEP1* associated with PAH. It also highlighted its possible involvement in PAH, which may be mediated in part by ceRNA interaction.

The 446 DEGs were obtained from GSE113439, a microarray database containing lung tissues from 11 control and 15 PAH patients. We obtained 199 immune-related DEGs from 446 DEGs by ImmPort analysis. Among them, ITGAL and other genes related to NETs were identified; moreover, NETs have a determining role in PAH through the activation of platelets and endothelial cells [20]. The immune-related ceRNA network is based on prediction-validated miRNA-mRNA/lncRNA regulation and its associated levels. In this ceRNA network, the lncRNA and mRNA shared at least one miRNA and the expression of lncRNA and mRNA |Pearson’s correlation| > 0.8. Through the above analysis, we obtained 203 lncRNA-mRNA pairs. Among them, typical immune-related mRNAs including *TBK1*, *IREB2* and *RABEP1* were screened. 

There are seven candidate lncRNAs (*HOXA-AS2*, *JPX*, *LMNTD2-AS1*, *MALAT1*, *MIR99AHG*, *SND1-IT1* and *ZNF22-AS1*) for the 203 lncRNA-mRNA pairs in the ceRNA network that were found to be associated with PAH for the first time. A molecular switch called *JPX* renders the X chromosome inactive [21,22]. Relevant investigations have revealed that although the nucleotide sequence and RNA secondary structure of human *JPX* and its mouse lncRNA *JPX* homologue are quite different, both lncRNAs exhibit significant binding to CTCF, and human *JPX* can functionally compensate for *JPX* deletion in mice embryonic stem cells [23]. *JPX* has recently been linked to reports of lung cancer. *JPX* is a major oncogene that is elevated in NSCLC tissues and is linked to a poor prognosis [24]; *JPX* interacts with miR-145-5p to increase cyclin D2 expression in a ceRNA pathway, promoting the growth and development of NSCLC [25]. Metastasis-associated lung adenocarcinoma transcript 1 (*MALAT1*), also known as nuclear enriched transcript 2 (*NEAT2*), is a lncRNA that has received a lot of interest recently [26,27,28]. As its name suggests, it increases cancer cell metastasis and is hence associated with higher overall metastatic potential in various malignancies [29], although some studies have even suggested its metastasis suppressor activity [30]. In this study, we chose a higher |Pearson’s correlation coefficient| > 0.91 for the lncRNA-mRNA pairs screening, and such screening conditions ensured the higher reliability of the results. Through this analysis, we obtained lncRNA-mRNA pairs including three lncRNAs and 15 mRNAs. Among these lncRNA-mRNA pairs, *SND1-IT1* was cancelled due to the absence of associated lncRNAs, and also because all the lncRNA-mRNAs we obtained were positively correlated with PAH, only two lncRNAs (*JPX* and *MALAT1*) remained. Notably, in an animal PAH model, there was no significant difference in *malat1* in the lung between the control and PAH group. However, there were significant differences between the two groups in cardiac tissue, which may be because *malat1* is highly expressed in cardiac tissue, which also indicates that PAH causes heart disease. It has been shown that *MALAT1* silencing elevated miR-26a-5p to protect against sepsis-induced myocardial injury by reducing *rcan2* [31]. At the same time, *Malat1*-mediated intramitochondrial homeostasis enhances cardiac microcirculation resistance to hypoxic injury, thereby improving prognosis in MI mice [32]. These results indicate that *MATAT1* is more likely to be a lncRNA mediating cardiomyopathy. In addition, our co-expression analysis showed that these lncRNAs were strongly correlated with immune-related genes such as *HMGB1*, *TBK1*, *IREB2* and *RABEP1*. We speculate that these two candidate lncRNAs affect immune-related genes through miRNAs, thereby affecting immune status, and ultimately affecting the occurrence and development of PAH.

Many immune-related lncRNA-mRNA pairs were found through co-expression analyses, but only one pair (*JPX-RABEP1*) was verified by another dataset and experiment, which showed the accuracy as well as importance of this pair. *Rabaptin*, RAB GTPase binding effector protein 1 (*RABEP1*) also known as *Rbpt5*, is involved in vesicle-mediated transport and located in the endocytic vesicle and endosome [33]. It is a long-established regulator of early endosome function in targeting the autophagy machinery to early endosomes damaged by chloroquine [34]. It is meaningful to be the first study identifying the novel immune-related lncRNA-mRNA pair *JPX-RABEP1* associated with PAH. However, only *JPX-RABEP1* is confirmed in the animal model, despite the fact that all lncRNA-mRNA pairs achieve consistent results in both the training and validation datasets. First off, the fact that our training and validation datasets were from several studies shows how general the lncRNA-mRNA combinations were across diverse ethnic groups. Second, it is acceptable that not all lncRNA-mRNA pairs could be verified in the mouse PAH model because the animal model still has certain limitations when it comes to simulating the etiology and mechanism of human PAH. Meanwhile, there are numerous approaches, including the subcutaneous injection of monocrotaline, hypoxia and surgical shunt models. We cannot confirm whether the lncRNA-mRNA pair (*JPX-RABEP1*) produced from the validation of various PAH modeling modalities is the same; all we can show is that we were able to generate the *JPX-RABEP1* pair using the animal model of hypoxia with SU5416 (20 mg/kg). In fact, these results also support the above trials’ findings that *JPX-RABEP1* pairs were crucial in the emergence of PAH. Because this study only examined the relationship between the lncRNA and mRNA as well as the lncRNA-mRNA pair and PAH, more research is required to determine the precise intervention mechanisms.

The results of an immune infiltration analysis showed that the proportion of several immune cells were dysregulated in PAH. The strong correlation between the expression of the *JPX-RABEP1* pair and the dysregulated immune cells indicated that the *JPX-RABEP1* pair may affect the occurrence and development of PAH through immune cell infiltration.

The following limitations exist in this study: (1) increasing the sample size and improving the genetic information may improve the accuracy of disease assessment and prediction; (2) the expression of rabep1 should be detected by overexpressing or knocking down JPX in cell lines, and then exploring the molecular mechanism; (3) the PAH model in mice cannot fully imitate the human PAH process, and human lung and heart clinical samples need further study to verify the relationship between immune-related lncRNA-mRNA pairs and PAH.

## 4. Materials and Methods

### 4.1. Data Source

The microarray datasets GSE113439 and GSE53408 were sourced from the GEO database (http://www.ncbi.nlm.nih.gov/geo/ (accessed on 26 October 2021)). The workflow for bioinformatics analysis in our study is illustrated (Figure 8).

### 4.2. Data Preprocessing and DEGs Screening

A pipeline of HISAT2 [35], Samtools [36] and featureCounts [37] was used for aligning the trimmed reads to the human reference genome (GRCh38) and quantifying gene expression. Only uniquely mapped reads were used for expression quantification. According to the workflow, the GSE113439 dataset served as a training dataset to identify important genes while the GSE53408 dataset was used for validation. All data were statistically analyzed and visualized using R software 4.1.2 and relevant packages. DEGs in the GSE113439 dataset were screened with *p* value < 0.05 and |Log2 fold change| > 1 as the empirical analysis cutoff by the ‘DESeq2′ package.

### 4.3. Identification of Immune-Related lncRNAs

A list of immune-related genes was downloaded from the gene list resources in the Immunology Database and Analysis Portal (ImmPort) (https://www.immport.org/ (accessed on 1 June 2022)). The intersection of the above immune-related genes and genes of DEGs was obtained and defined as immune-related mRNAs. Immune-related lncRNAs were screened by the co-expression analysis of immune-related mRNAs as described above.

### 4.4. Construction and Analysis of an Immune-Related lncRNA-Associated Competing Endogenous RNA Network

Immune-related lncRNAs and mRNAs were used to construct a competing endogenous RNA (ceRNA) network. The miRcode database was used to collect predicted and experimentally validated microRNA (miRNA)-mRNA-interaction data, as well as miRNA-lncRNA-interaction data. Competing lncRNA-mRNA pairs were identified using the following criteria: (1) the lncRNA and mRNA must share miRNAs; and (2) the expression of lncRNA and mRNA must be positively correlated (|Pearson’s correlation| > 0.8). These identified lncRNA-mRNA pairs were used to construct the ceRNA network and were visualized using Cytoscape 3.9.1 software.

### 4.5. Immune Cell Infiltration Analysis

To quantify the relative proportions of infiltrating immune cells in PAH, CIBERSORT, a method of analysis of the different immune cell types to tissues, was adopted to analyze the merged expression data and calculate immune cell infiltrations. A *p* value < 0.05 was used to filter the samples. The percentage of each immune cell type in the samples was calculated and displayed in a bar plot. The heat map of the 22 immune cells was made using the ‘pheatmap’ package. The ‘vioplot’ package was used to compare and visualize the levels of 22 immune cells between the PAH and control samples. 

### 4.6. Animal-PAH Model and Samples

The PAH mouse model was constructed by Dr. Jin Tingting of Run Run Shaw Hospital. The lung and heart mouse PAH samples were also provided by her. Briefly, male mice aged 6–8 weeks were selected as animal models, and the PAH model was induced by hypoxia combined with the subcutaneous injection of SU5416 (20 mg/kg) once a week for a total of four weeks. The normal group was placed in the same room and raised for 4 weeks in a normal pressure and normal oxygen environment, and the feeding conditions remained basically the same.

### 4.7. Histopathologic Evaluation of Lung Tissues

For histological analysis, the lung tissues were fixed in a fresh 4% formaldehyde solution for 24 h and were then dehydrated, transparent, dipped in wax, and embedded in paraffin. Finally, 5-μm sections were cut and stained with hematoxylin-eosin. The tissue sections were observed under a light microscope to examine the lung histopathology.

### 4.8. Quantitative Real-Time PCR (qRT-PCR)

Total RNA was extracted as previously described [38]. In short, total RNA of the cultured tissues was extracted using TRIZol reagent (Tsingke, Beijing, China). The HiScript III RT SuperMix for qPCR (Vazyme, Nanjing, China) was performed to synthesize the cDNA with 1 μg of RNA. Next, each 96-well plate well was mixed with 6.5 μL of cDNA (diluted at 1:20), 7.5 μL of qPCR SYBR Master Mix (Vazyme, Nanjing, China) and 1 μL of primers (10 mM). The primers used in the study are listed in Appendix A.

### 4.9. Statistical Analysis

R software was used to evaluate the correlation between the lncRNAs and mRNAs. A T-test was used to analyze the difference of RNA expression levels between the control group and PAH group. A Kruskal-test was used to analyze the difference of gene expression levels as well as the immune cell composition in the microarray dataset between the control group and the PAH group. The statistical significance was set at a *p* value < 0.05.

## 5. Conclusions

Through the use of a microarray dataset, the expression profiles of mRNA and lncRNA in lung tissues from PAH patients were examined in our study. We discovered a pair of immune-related lncRNAs and mRNAs that shared regulatory miRNAs and had a strong association, consisting of two lncRNAs and six mRNAs. Additionally, one lncRNA-mRNA pair (*JPX-RABEP1*) was validated using an animal model and validating dataset. Additionally, findings from an investigation into immune cell infiltration suggested that JPX-RABEP1 may have a role in the emergence and progression of PAH via immune cell infiltration. The lncRNA-mRNA pair *JPX-RABEP1* may be a new biomarker and therapeutic target for PAH, according to the study’s findings.

## Figures and Tables

**Figure 1 ijms-23-15559-f001:**
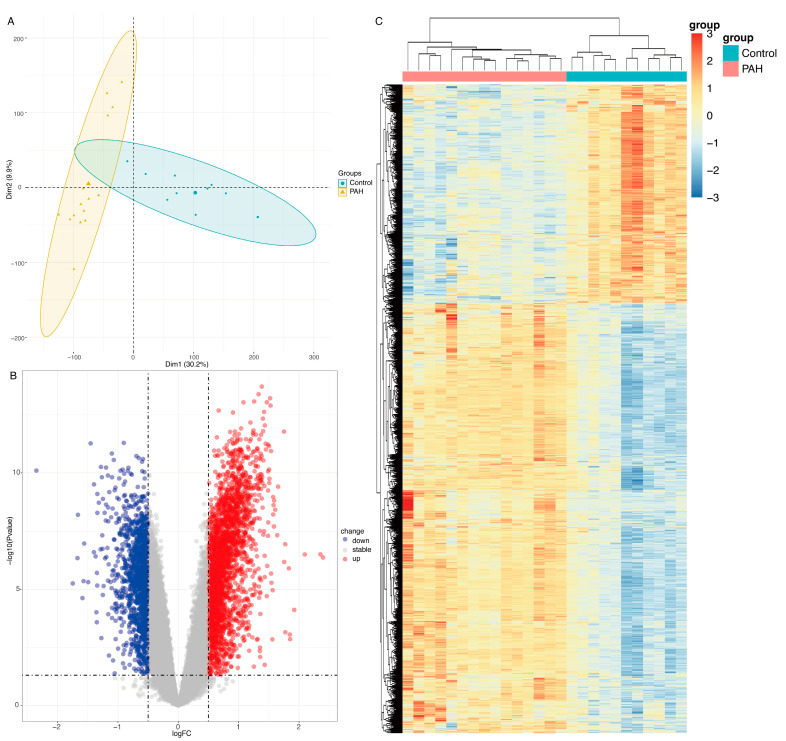
PCA, volcano plot and heatmap for DEGs identified in the GSE113439 dataset. (**A**) PCA plots. Blue dots represent control patients and yellow triangles represent PAH patients. (**B**) Volcano plot of DEGs (*p*-values < 0.05 and |Log2 fold change| > 1) identified through the GSE113439 dataset. Each dot represents an individual gene. Red dots denote upregulated DEGs, while blue dots represent downregulated DEGs. (**C**) Heatmap of all DEGs. Red blocks represent upregulated DEGs and blue blocks represent downregulated DEGs in those groups. PAH, lung tissues of pulmonary arterial hypertension patients; Control, normal lung tissue; DEGs, differentially expressed genes.

**Figure 2 ijms-23-15559-f002:**
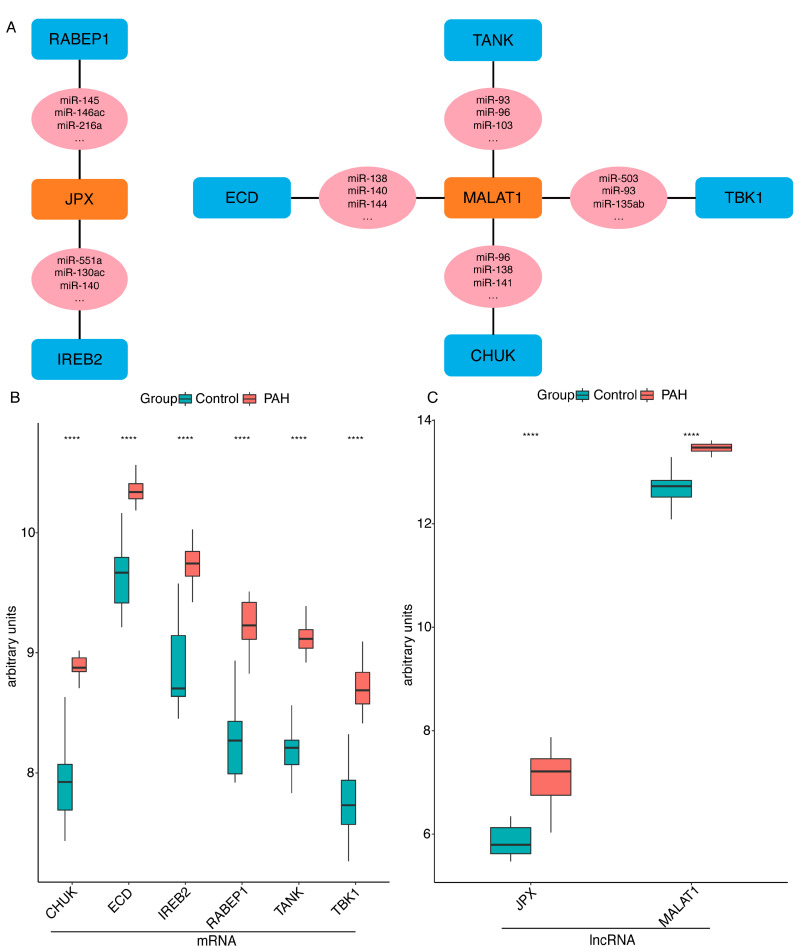
The immune-related ceRNA network and the expression of lncRNA-mRNA pairs in the training GSE113439 dataset. (**A**) The immune-related ceRNA network. The blue nodes represent mRNAs, the pink nodes represent miRNAs and the orange nodes show lncRNAs. The mRNAs and lncRNA are linked by a common miRNA with a black thread. (**B**) The expression of mRNAs included in ceRNA network among the PAH and control samples. (**C**) Expression of lncRNAs included in ceRNA network among the PAH and control samples. PAH, pulmonary arterial hypertension; **** *p* value < 0.0001.

**Figure 3 ijms-23-15559-f003:**
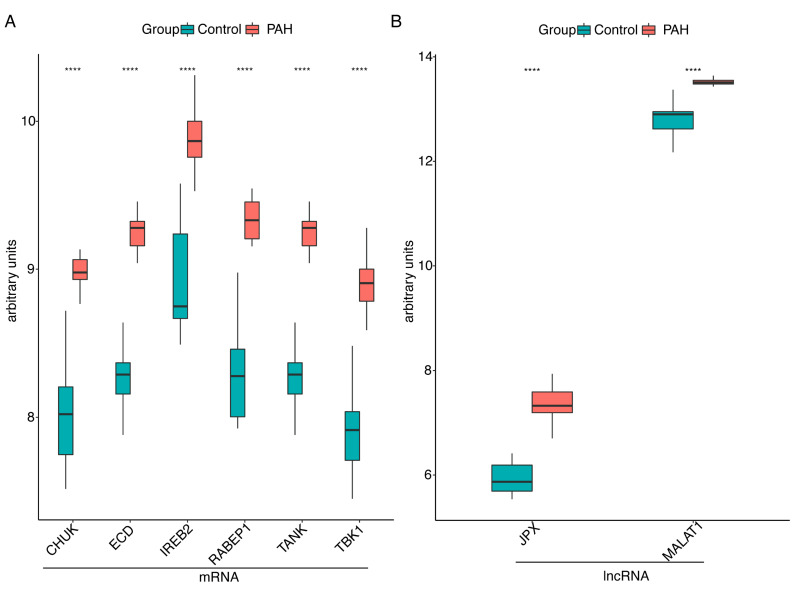
The expression of lncRNA-mRNA pairs in the training GSE53408 dataset. (**A**) Expression of mRNAs included in the ceRNA network among the PAH and control samples. (**B**) Expression of lncRNAs included in the ceRNA network among the PAH and control samples. PAH, pulmonary arterial hypertension; **** *p* value < 0.0001.

**Figure 4 ijms-23-15559-f004:**
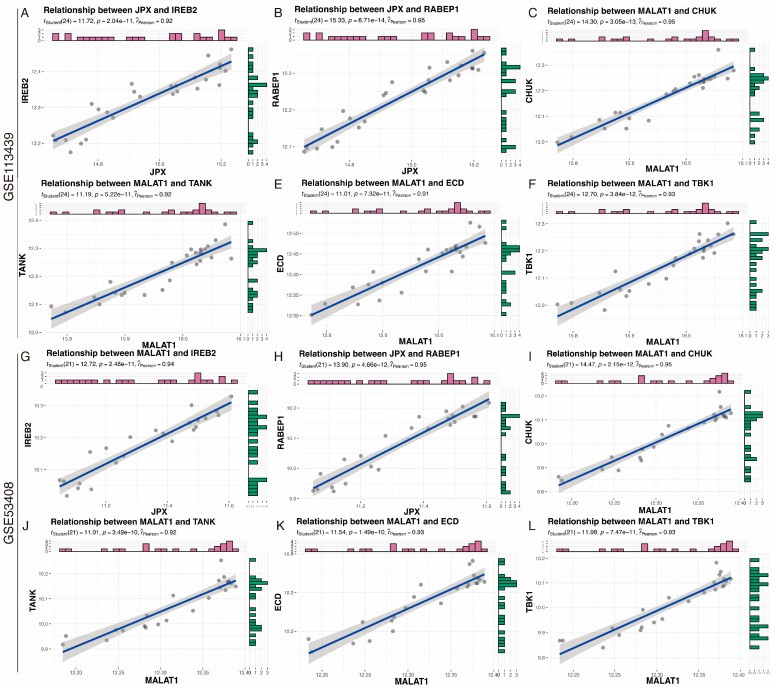
Validation of the correlation of immune-related lncRNA-mRNA pairs. (**A**–**F**) Correlation of immune-associated lncRNAs-mRNAs pairs in the GSE113439 dataset. (**G**–**L**) Validation of correlations of immune-related lncRNA-mRNA pairs in the validating GSE53408 dataset.

**Figure 5 ijms-23-15559-f005:**
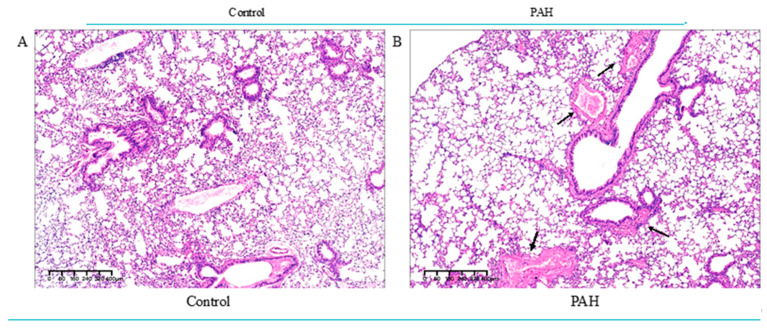
Histopathological sections of mouse lung tissue. (**A**) control group (**B**) PAH group. Histomorphological and pathological findings showed thickened vessel walls and increased alveolar fusion in the lungs of PAH mice, with significant lesions in the lung tissue. The arrows indicate the thickening of the pulmonary arteriole wall, lumen stenosis, and proliferation of vascular smooth muscle cells.

**Figure 6 ijms-23-15559-f006:**
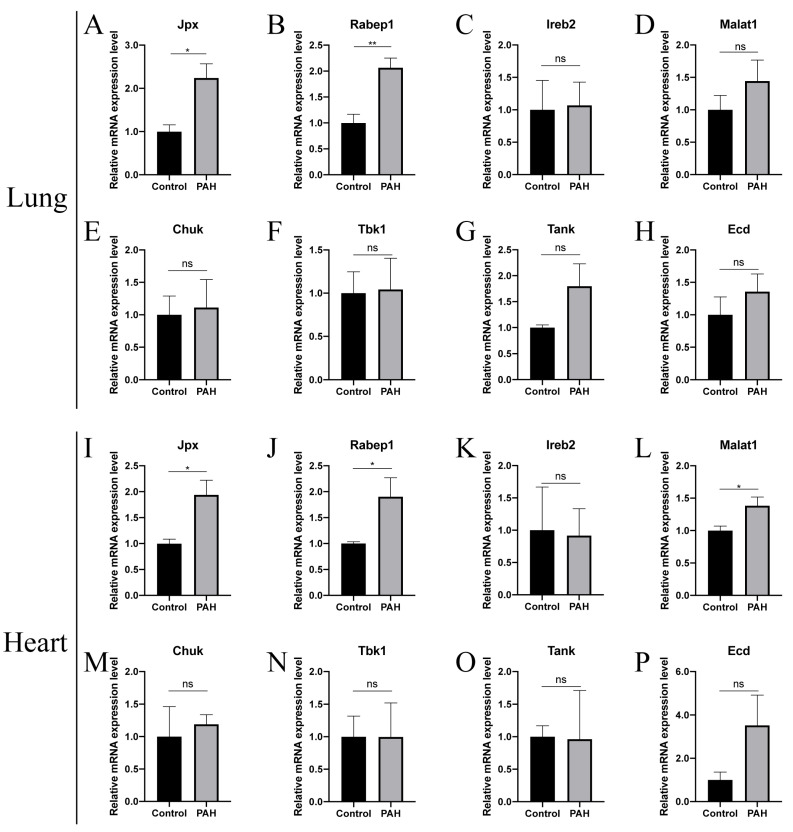
Expression of lncRNA-mRNA pairs in animal model. (**A**–**H**) Expression of lncRNA-mRNA pairs in lung tissues. (**I**–**P**) Expression of lncRNA-mRNA pairs in heart tissues. PAH, Pulmonary arterial hypertension; Control, normal tissues; ns, no significance. * *p* value < 0.05, ** *p* value < 0.01.

**Figure 7 ijms-23-15559-f007:**
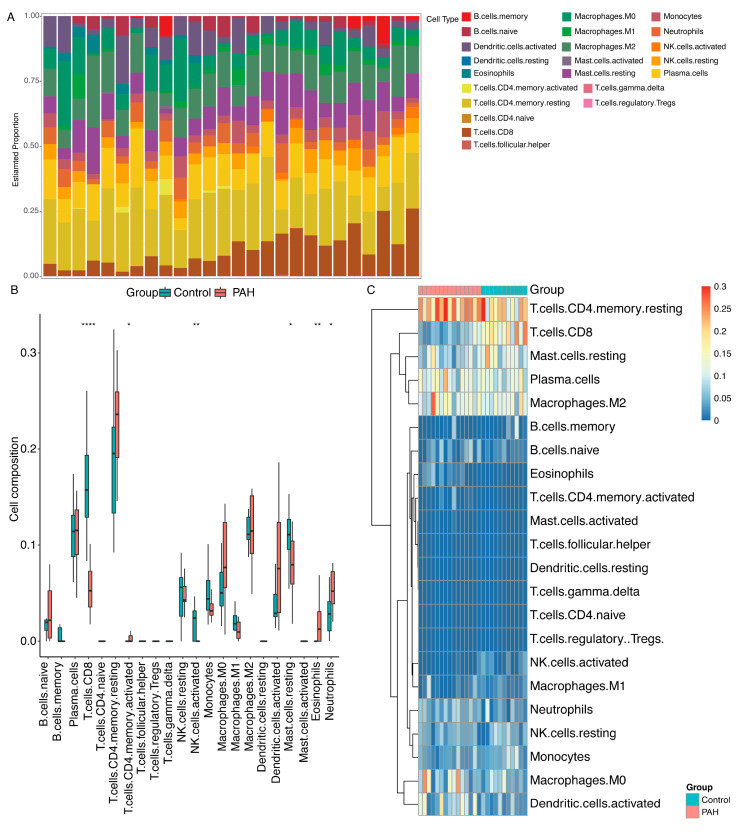
Immune cell infiltration in PAH and control lung tissues in the GSE113439 dataset. (**A**) The composition of 22 kinds of immune cells in each sample by histogram. (**B**) The composition of 22 kinds of immune cells in each sample by heatmap. (**C**) Comparison of 22 immune cell subtypes between patients in PAH and controls. Blue and red colors on behalf of control and PAH samples, respectively. * *p* value < 0.05, ** *p* value < 0.01, **** *p* value < 0.0001.

**Figure 8 ijms-23-15559-f008:**
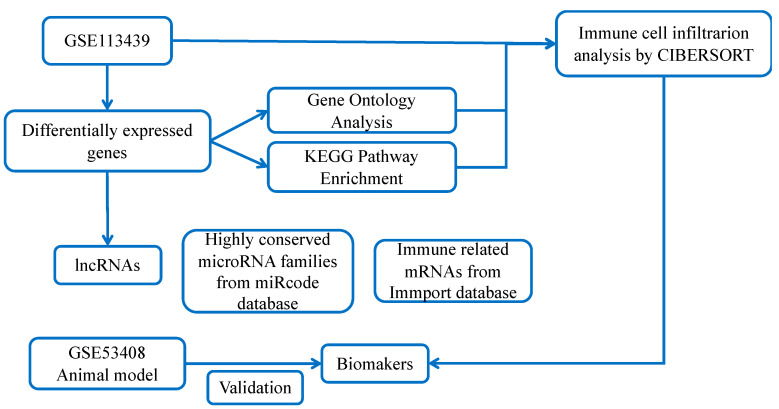
Study flowchart.

**Table 1 ijms-23-15559-t001:** The lncRNA-mRNA pairs with |correlation co-efficiency| > 0.91 in the co-expression analysis.

Immune-Related lncRNA	Immune-Related mRNA	Correlation Co-Efficiency
JPX	RABEP1	0.952570119
MALAT1	CHUK	0.946039227
SND1-IT1	RXRB	−0.943270822
MALAT1	MAPK3	−0.937237966
SND1-IT1	MAPK3	−0.933509908
MALAT1	TBK1	0.932978885
JPX	ICAM2	−0.931467245
MALAT1	TNFRSF14	−0.931453188
JPX	IREB2	0.92261343
JPX	GDF10	−0.921912199
SND1-IT1	CTF1	−0.921173014
SND1-IT1	ECD	0.920122958
SND1-IT1	HSPA5	0.920074041
MALAT1	NR2F1	−0.919527008
MALAT1	TANK	0.916072818
MALAT1	CTF1	−0.91363069
MALAT1	ECD	0.913586674
MALAT1	CMTM8	−0.910914511

## Data Availability

All figures and data used to support to this study are induced within this article.

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
