# Peer review of "Identification of JPX-RABEP1 Pair as an Immune-Related Biomarker and Therapeutic Target in Pulmonary Arterial Hypertension by Bioinformatics and Experimental Analyses"

_ijms, 2022, doi:10.3390/ijms232415559_

Round 1

Reviewer 1 Report

The present manuscript of Gong et al. analyzed the expression and a potential regulatory network of immune-related mRNAs, miRNAs and lncRNAs in samples from pulmonary arterial hypertension patients.

The construction and analysis of the immune-related ceRNA network and the immune cell infiltration analysis are described very detailed and understandable, also the study flowchart improves the understanding of the reader. I also appreciate that the authors list the limitations of the study and give an outlook on further experiments. However, the interpretation of the results is only rudimentary. The study concentrates on a bioinformatic analysis of existing microarray data. From my point of view this manuscript should rather be submitted to a journal that is more specialized to bioinformatic analyses. Below you can find additional concerns.

- The Figures (especially Figure 1, 5, 8, but also other figures) are too small and should be increased in size and sometimes also in quality (Figure 5). It was not possible to review figure 5.

- The H&E staining of mouse lungs does not show the presence of JPX-RABEP1. I would recommend to perform in situ hybridization experiments.

- ll. 90-91: “…while down-regulated DEGs are mainly enriched in negative regulation of megakaryocyte differentiation and response to hypoxia and other biological processes.” Such statements are highly imprecise. Please describe the results more accurately. In a plot of GO terms of biological processes, only biological processes can be shown. Moreover, was is the further meaning of these results?

- ll. 111-112: “A list of immune-related genes was downloaded from the gene list resources in Immunology Database and Analysis Portal (ImmPort)” The authors should explain the reason why they searched for immune related genes.

Author Response

The present manuscript of Gong et al. analyzed the expression and a potential regulatory network of immune-related mRNAs, miRNAs and lncRNAs in samples from pulmonary arterial hypertension patients.

The construction and analysis of the immune-related ceRNA network and the immune cell infiltration analysis are described very detailed and understandable, also the study flowchart improves the understanding of the reader. I also appreciate that the authors list the limitations of the study and give an outlook on further experiments. However, the interpretation of the results is only rudimentary. The study concentrates on a bioinformatic analysis of existing microarray data. From my point of view this manuscript should rather be submitted to a journal that is more specialized to bioinformatic analyses. Below you can find additional concerns.

- The Figures (especially Figure 1, 5, 8, but also other figures) are too small and should be increased in size and sometimes also in quality (Figure 5). It was not possible to review figure 5.

RE: Thanks for the reviewer’s suggestion. In order to make the picture clearer, we used high quality figures to replace previous figures, shown as figure 1, 4 and 7 in the revised manuscript.

- The H&E staining of mouse lungs does not show the presence of JPX-RABEP1. I would recommend to perform in situ hybridization experiments.

RE: Thanks for your constructive comment. In situ hybridization is a good technique to show the location of RNA expression, but the method requires more time and better technics. Based on our current platform, and time limitations, we are unable to obtain good in situ hybridization results. However, in our manuscript, we have validated the expression of JPX-RABEP1 pair using qPCR in mouse heart and lung tissues, the results showed that the expression of JPX-RABEP1 pair was increased in PAH mouse heart and lung. In future work, we will use in situ hybridization to further explore in which cell the JPX-RABEP1 pair plays an important role in PAH.

- ll. 90-91: “…while down-regulated DEGs are mainly enriched in negative regulation of megakaryocyte differentiation and response to hypoxia and other biological processes.” Such statements are highly imprecise. Please describe the results more accurately. In a plot of GO terms of biological processes, only biological processes can be shown. Moreover, was is the further meaning of these results?

RE: Thanks for your suggestion. This part is a re-analysis of the original data to confirm whether it is consistent with the content of the previous paper [1], according to the suggestions of the reviewers, we removed this part and previous figure 2 in the revised manuscript.

- ll. 111-112: “A list of immune-related genes was downloaded from the gene list resources in Immunology Database and Analysis Portal (ImmPort)” The authors should explain the reason why they searched for immune related genes.

RE: Thanks for your suggestion. The pathogenesis of PAH is complicated, recently studies suggest that inflammatory response is the main pathogenic mechanism, therefore, we searched for immune-related genes in PAH by big-data analysis and validated them by mouse PAH model. The results showed that JPX-RABEP1 pair may participate in the occurrence and development of PAH through immune cell infiltration.

Thus, we added the sentences as below in the updated manuscript to explain the reason.

“The involvement of inflammation in the onset of the illness has recently come to light in preclinical and clinical PAH investigations. It was first noted that several inflammatory disorders, such as connective tissue illnesses, are linked to an elevated prevalence of PAH. Then, in lung samples from PAH patients, nearly all lineages of inflammatory cells—most notably macrophages, mast cells, T lymphocytes, B lymphocytes, dendritic cells, and neutrophils—were found close to the altered pulmonary vasculature [2].”

References:

  1. Li, A.; He, J.; Zhang, Z.; Jiang, S.; Gao, Y.; Pan, Y.; Wang, H.; Zhuang, L., Integrated Bioinformatics Analysis Reveals Marker Genes and Potential Therapeutic Targets for Pulmonary Arterial Hypertension. Genes (Basel) 2021, 12, (9).
  2. Hu, Y.; Chi, L.; Kuebler, W. M.; Goldenberg, N. M., Perivascular Inflammation in Pulmonary Arterial Hypertension. Cells 2020, 9, (11).

Reviewer 2 Report

1. What is the purpose of doing GO term analysis? Are the GO terms related to immune cell infiltration or other PAH pathology? It seems there was no conclusions derived from it for subsequent steps. It should be removed if this is the case.

2. In Fig 8, please state in the text and figure legend which GEO dataset was used for this analysis.

3. Is the PAH mouse model constructed by authors or was just the tissue obtained from elsewhere? The methods show that just the tissue was obtained from Run Run Shaw Hospital. Please clarify in the results section whether the mouse was made by the authors or not.

Numerous studies have previously shown via bioinformatics analysis that there is a significant increase in certain immune cell types in PAH. If the PAH mouse model is indeed available to the authors, can immune cell infiltration be studied in vivo?

Author Response

  1. What is the purpose of doing GO term analysis? Are the GO terms related to immune cell infiltration or other PAH pathology? It seems there was no conclusions derived from it for subsequent steps. It should be removed if this is the case.

RE: Thanks for the reviewer’s suggestion. This part is a re-analysis of the original data to confirm whether it is consistent with the content of the previous paper [1], according to the suggestions of the reviewers, we removed this part and previous figure 2 in the revised manuscript.

  1. In Fig 8, please state in the text and figure legend which GEO dataset was used for this analysis.

RE: Thank you for pointing out the mistake. We didn’t mention it clearly before, we have revised this part in the updated manuscript.

  1. Is the PAH mouse model constructed by authors or was just the tissue obtained from elsewhere? The methods show that just the tissue was obtained from Run Run Shaw Hospital. Please clarify in the results section whether the mouse was made by the authors or not.

RE: Thanks for your constructive comment. PAH mouse model was constructed by Dr. Tingting Jin from Run Run Shaw Hospital and the tissue sample used in this manuscript is also provided by Dr. Tingting Jin. The related experiments were done by the authors.

Meanwhile, we updated “To further verify the reliability of the obtained lncRNA-mRNA pairs in PAH, lung and heart samples were collected from mouse model of PAH (gifted by Dr. Tingting Jin).” in the revised manuscript.

Numerous studies have previously shown via bioinformatics analysis that there is a significant increase in certain immune cell types in PAH. If the PAH mouse model is indeed available to the authors, can immune cell infiltration be studied in vivo?

RE: Thanks for your suggestion. In this study, we aimed to analyze immune related biomarkers in PAH through bioinformatics. Here, we identified that JPX-RABEP1 pair is a potential biomarker for PAH. In the future study, we will use the mouse model to verify the role of immune cell infiltration and JPX-RABEP1 pair in the occurrence and development of PAH in vivo.

References:

  1. Li, A.; He, J.; Zhang, Z.; Jiang, S.; Gao, Y.; Pan, Y.; Wang, H.; Zhuang, L., Integrated Bioinformatics Analysis Reveals Marker Genes and Potential Therapeutic Targets for Pulmonary Arterial Hypertension. Genes (Basel) 2021, 12, (9).

Reviewer 3 Report

In the manuscript submitted to IJMS entitled “Identification of JPX-RABEP1 Pair as an Immune-related Biomarker and Therapeutic Target in Pulmonary Arterial Hypertension (PH) by Bioinformatics and Experimental Analyses”, the authors studied immune-related lncRNA and their target mRNA by analyzing public databases and mouse models and identified JPX-RABEP1 as a therapeutic target for pulmonary arterial hypertension. PH is an important vascular disease and the analyses would help understand the disease's pathophysiology.

The reviewer has several suggestions for improvement.

On page 2, lines 56-59, Ref [15] is not the authors’ study; however, the authors' sentences seem to mean that could be the authors’ study. The reviewer understands that the authors used the database; however, the reviewer advises correcting the sentence for clarity.

On page 2, line 65, it is advisable to specify which miRNA(s) the authors mean.

NETs are important for inflammation and whether they may be involved in PH pathophysiology should be described more.

In Figure 3, the authors stated “the red nodes on behalf of miRNAs”; however, the color seems pink, but not red.

Line 165 on page 6, GSE53408 (Figure 5E-L) should read GSE53408 (Figure 5G-L).

In Figure 6, it is better to indicate the thickening of the pulmonary arteriole wall, lumen stenosis, and proliferation of vascular smooth muscle cells by arrows and/or arrowheads.

On page 8 line 183, Figure 7A-C should be Figure 7A-H, and Figure 7D-F should be Figure 7I-P.

The discussion section is too long.

LncRNA is used in several occasions, in the case where lncRNA is suitable.

In references, unify the description of page numbers. For example, in Ref 1, 306-22, whereas in ref 3, it is shown as 1735-1767.

Author Response

In the manuscript submitted to IJMS entitled “Identification of JPX-RABEP1 Pair as an Immune-related Biomarker and Therapeutic Target in Pulmonary Arterial Hypertension (PH) by Bioinformatics and Experimental Analyses”, the authors studied immune-related lncRNA and their target mRNA by analyzing public databases and mouse models and identified JPX-RABEP1 as a therapeutic target for pulmonary arterial hypertension. PH is an important vascular disease and the analyses would help understand the disease's pathophysiology.

The reviewer has several suggestions for improvement.

On page 2, lines 56-59, Ref [15] is not the authors’ study; however, the authors' sentences seem to mean that could be the authors’ study. The reviewer understands that the authors used the database; however, the reviewer advises correcting the sentence for clarity.

RE: Thanks for the reviewer’s suggestion. We have changed this sentence as below in the revised manuscript.

“In our study, we investigated the expression profiles of mRNAs and lncRNAs in the lung tissues by downloading publicly available microarray dataset from 11 control and 15 PAH patients (GSE113439) [15].”

On page 2, line 65, it is advisable to specify which miRNA(s) the authors mean.

RE: Thanks for your constructive comment. According to your suggestion, the shared miRNAs were added in the revised manuscript, lines 65-66.

NETs are important for inflammation and whether they may be involved in PH pathophysiology should be described more.

RE: Thanks for your comments. According to the suggestions of the other two reviewers, we deleted the GO and KEGG analysis charts, however, genes related to NETs, such as ITGAL and the relationship between NETs and PAH are added in the discussion section. The sentences as below were updated in the discussion part of the revised manuscript, lines 302-348.

“Among them, ITGAL and other genes related to NETs were identified, moreover, NETs have a determining role in PAH through activation of platelets and endothelial cells [1].”

In Figure 3, the authors stated “the red nodes on behalf of miRNAs”; however, the color seems pink, but not red.

RE: Thank you for pointing out the mistake. We have replaced red with pink.

Line 165 on page 6, GSE53408 (Figure 5E-L) should read GSE53408 (Figure 5G-L).

RE: Thank you for pointing out the mistake. We have revised this part.

In Figure 6, it is better to indicate the thickening of the pulmonary arteriole wall, lumen stenosis, and proliferation of vascular smooth muscle cells by arrows and/or arrowheads.

RE: Thanks for your suggestion. Using arrows and/or arrowheads to indicate the thickening of the pulmonary arteriole wall, lumen stenosis, and proliferation of vascular smooth muscle cells can indeed make the figure easier to be understand. The arrow shows the pathological changes in PAH in the figure 5.

Meanwhile, we updated “The arrows indicate the thickening of the pulmonary arteriole wall, lumen stenosis, and proliferation of vascular smooth muscle cells.” in results and figure legend, respectively.

On page 8 line 183, Figure 7A-C should be Figure 7A-H, and Figure 7D-F should be Figure 7I-P.

RE: Thank you for pointing out the mistake. We have revised this part in the updated manuscript.

The discussion section is too long.

RE: Thanks for your suggestion. We have simplified the discussion part in the revised manuscript.

LncRNA is used in several occasions, in the case where lncRNA is suitable.

RE: Thank you for pointing out the mistake. We have replaced all LncRNA with lncRNA in the updated manuscript.

In references, unify the description of page numbers. For example, in Ref 1, 306-22, whereas in ref 3, it is shown as 1735-1767.

RE: Thank you for pointing out the mistakes. We have updated the references in the revised manuscript.

References:

  1. Baptista de Barros Ribeiro Dourado LP, Santos M, Moreira-Gonçalves D. Nets, pulmonary arterial hypertension, and thrombo-inflammation. J Mol Med (Berl). 2022 May;100(5):713-722.

Round 2

Reviewer 1 Report

- Some figures have may increased in size but not in quality. The authors must increase the quality of the figures. From my point of view it is the absolute minimum to make sure that the reader can read the font and see every detail.

- Again I would recommend to perform more in vitro or in vivo experiments. Else this manuscript would fit better into a bioinformatic journal.

Author Response

- Some figures have may increased in size but not in quality. The authors must increase the quality of the figures. From my point of view it is the absolute minimum to make sure that the reader can read the font and see every detail.

RE: Thanks for the reviewer’s suggestion. To make the figures clearer, we increased the figures size and quality in the updated manuscript. Meanwhile, we have uploaded high resolution TIFF figures in the file

- Again I would recommend to perform more in vitro or in vivo experiments. Else this manuscript would fit better into a bioinformatic journal.

RE: Thanks for your suggestion. In this study, we explored for immune-related genes in PAH by big-data analysis and validated them by in vivo mouse model using qPCR and H&E staining experiments to confirm that JPX-RABEP1 pair is a potential biomarker for PAH.

We believe that our manuscript is in line with the theme of the IJMS Special Issue "Novel Therapeutic Targets for Pulmonary Arterial Hypertension" and sincerely hope that our findings may provide new therapeutic targets for PAH.